# FUSQA: Fetal Ultrasound Segmentation Quality Assessment

**Sevim Cengiz**                                          SEVIM_CENGIZ@ICLOUD.COM

**Ibrahim Almakky**                                    IBRAHIM.ALMAKKY@MBZUAI.AC.AE

**Mohammad Yaqub**                                MOHAMMAD.YAQUB@MBZUAI.AC.AE

*Computer Vision Department, MBZUAI, Abu Dhabi, UAE*

**Editors:** Accepted for publication at MIDL 2023

## Abstract

Deep learning models have been effective for various fetal ultrasound segmentation tasks. However, generalization to new unseen data has raised questions about their effectiveness for clinical adoption. Normally, a transition to new unseen data requires time-consuming and costly quality assurance processes to validate the segmentation performance post-transition. Segmentation quality assessment efforts have focused on natural images, where the problem has been typically formulated as a dice score regression task. In this paper, we propose a simplified Fetal Ultrasound Segmentation Quality Assessment (FUSQA) model to tackle the segmentation quality assessment when no masks exist to compare with. We formulate the segmentation quality assessment process as an automated classification task to distinguish between good and poor quality segmentation masks for more accurate gestational age estimation. We validate the performance of our proposed approach on two datasets we collect from two hospitals using different ultrasound machines. We compare different architectures, with our best-performing architecture achieving over 90% classification accuracy on distinguishing between good and poor quality segmentation masks from an unseen dataset. Additionally, there was only a 1.45-days difference between the gestational age reported by doctors and estimated based on CRL measurements using well-segmented masks. On the other hand, this difference increased and reached up to 7.73 days when we calculated CRL from the poorly segmented masks. As a result, AI-based approaches can potentially aid fetal ultrasound segmentation quality assessment and might detect poor segmentation in real-time screening in the future.

**Keywords:** Segmentation quality assessment, fetal ultrasound, gestational age estimation, and deep learning.

## 1. Introduction

Monitoring fetal growth, especially at the early stages of pregnancy, between 12 and 14 weeks, has proven to be an effective approach to detect possible fetal anomalies. More specifically, 5 out of 7 anomalies were found before 20 gestation weeks (Drysdale et al., 2002). Ultrasound is safe, non-invasive imaging modality which is widely used to monitor fetal growth. Real-time ultrasound allow sonographers to measure the crown-rump length (CRL) at the Dating scan, which is obtained from the crown of the fetal head to the bottom of the fetal rump. The measured CRL is then used to calculate Gestational Age (GA), which helps assess fetal growth against a standardized fetal growth chart. Inaccurate CRL measurement could lead to incorrect GA calculation, resulting in a wrong assessment of fetal growth. This highlights the importance of precise CRL measurement to detect potential fetal anomalies. CRL measurements can be either obtained manually by the sonographer

or by segmentation-based algorithms (Cengiz and Yaqub, 2021). The manual process of placing calipers on the fetal head and rump might be challenging for an inexperienced sonographer, especially if the image is acquired in a partially incorrect view.

Automatic algorithms have been developed to obtain measurements from fetal ultrasound images (Yaqub et al., 2013; Płotka et al., 2021). They typically depend on semantic segmentation of fetal anatomical structures such as fetal head, body, and palate (Ryou et al., 2019; Cengiz and Yaqub, 2021). The accuracy of the CRL measurement requires precise fetal segmentation from ultrasound images that adhere to clinical guidelines. Segmentation models are trained on ultrasound images along with annotated masks and their performance is assessed before they are considered for clinical adoption. However, challenges appear when applying such segmentation models to unseen data especially scans from different data sources e.g., machines, hospitals, population, etc. This requires quality assurance processes to check the quality of the segmentation to preserve the CRL measurement performance. Normally, this is done using a manual verification approach by sampling and annotating a subset of new unseen data, which is a costly and time-consuming task. In addition, the manual check for the quality of the segmentation must continue indefinitely to ensure proper results from any automatic segmentation model. Therefore, it is important to develop solutions which can automatically verify the quality of the segmentation without (or with minimal) manual intervention. In this work, we develop an automated Fetal Ultrasound Segmentation Quality Assessment (FUSQA) approach to obviate this manual verification process through a model that can distinguish between good and poor segmentation predictions on unseen data. More specifically, the contributions of this work are:

- We propose a simplified deep-learning-based method for automatic Fetal Ultrasound Segmentation Quality Assessment (FUSQA) to verify segmentation quality. We then compare this simplified approach to other deep-learning-based approaches such as Siamese and Synergic models, as well as other state-of-the-art methods.

- We conduct a two-site study to demonstrate the effectiveness of our proposed approach on unseen data acquired from different ultrasound machines.

- We demonstrate the importance of our automated fetal ultrasound image quality assessment approach on a clinical essential downstream task (accurate CRL measurement and GA estimation).

## 2. Related Work

In this section, we review the approaches that focused on the fetal ultrasound segmentation task as well as segmentation and detection quality assessment.

**Fetal Ultrasound Segmentation.** There has been significant work done on segmentation from fetal ultrasound images. These efforts focused either on enabling better structure visualization, measuring structure lengths, area, or volume, and assessing calculated metrics against clinical guidelines. To accomplish this, the focus has been on deep-learning-based segmentation methods to segment fetal anatomical structures such as placenta (Looney et al., 2018; Yang et al., 2019; Zimmer et al., 2019; Schwartz et al., 2022), gestational sac (Yang et al., 2019), fetal CRL (Ryou et al., 2016, 2019; Cengiz and Yaqub, 2021), fetal heart

(Philip et al., 2019; Nurmaini et al., 2021), femur (Zhu et al., 2021), abdomen (Ravishankar et al., 2016), fetal head for head circumference (Yaqub et al., 2013; Sobhaninia et al., 2019; Maraci et al., 2020).

**Segmentation and Detection Quality Assessment.** It is challenging to guarantee the performance of automatic segmentation and detection models, especially for real-time applications or when deployed on unseen data. Therefore, the segmentation quality assessment task is designed to predict the performance of segmentation models in such scenarios. Most of the work on segmentation quality assessment has been conducted on natural images (Zhou et al., 2020; Asgari Taghanaki et al., 2021; Chen et al., 2018), but some work has been done on medical images (Zhao et al., 2022; Kohlberger et al., 2012; Sunoqrot et al., 2020; Budd et al., 2019), with a majority focused on cardiovascular MR images.

Robinson et al. conducted a study for real-time automated quality control for cardiovascular MR segmentation (Robinson et al., 2018). Puyol-Anton et al. (Puyol-Antón et al., 2020) also addressed the quality-control process for cardiovascular MR images using Convolutional Neural Networks (CNNs) and Bayesian inference for uncertainty-based quality control mechanism. On the other hand, Ruijsink et al. (Ruijsink et al., 2020) developed a two-step quality control mechanism for cardiac function analysis from magnetic resonance before and after segmentation. The pre-segmentation process is focused on validating image quality while the post-segmentation process uses a Support Vector Machine to detect abnormalities. Galati and Zuluaga (Galati and Zuluaga, 2021) formulated the segmentation quality assessment problem as an anomaly detection problem, where they trained a Convolutional Autoencoder (CAE) model to learn the variability of cardiac segmentation masks. Wu et al. (Wu et al., 2017) focused on quality control for fetal ultrasound, where they developed their Fetal Ultrasound Image Quality Assessment (FUIQA) method to decrease the measurement error caused by different sonographers. FUIQA employs a CNN model to find the region of interest of the fetal abdominal region in the ultrasound image. Another CNN model is then used to assess the image quality by assessing how well depicted are the key stomach structures. As such, FUIQA's second CNN is focused on quality control for the detection task performed by the first CNN model. However, in this work, we focus on the segmentation quality assessment process toward more accurate CRL measurement and the subsequent accurate gestation estimation on previously unseen data. To the best of our knowledge, this is the first work to address the segmentation quality assessment task in fetal ultrasound images for CRL measurement.

## 3. Proposed Method

In this section, we will describe our proposed simplified deep-learning-based method for automatic fetal ultrasound segmentation quality assessment. Starting from a set of ultrasound images and segmentation masks $\{(x_1, y_1), \ldots, (x_n, y_n)\} \in D_A$, which we use to train a segmentation model $S_A(x) = \hat{y}$, where $\hat{y} \approx y$. The difference between $\hat{y}$ and a dilated $\hat{y}$ mask using a kernel size of $3 \times 3$ provides a contour, which we then employ to get $\Delta_{CRL}$, the longest distance for the CRL measurement. Employing the work done by (Papageorghiou et al., 2014), we then calculate an estimated GA ($\hat{g}$). As such the overall aim is to predict a segmentation mask $\hat{y}$ that provides $\hat{g}$ that is as close as possible to the actual GA ($g$).

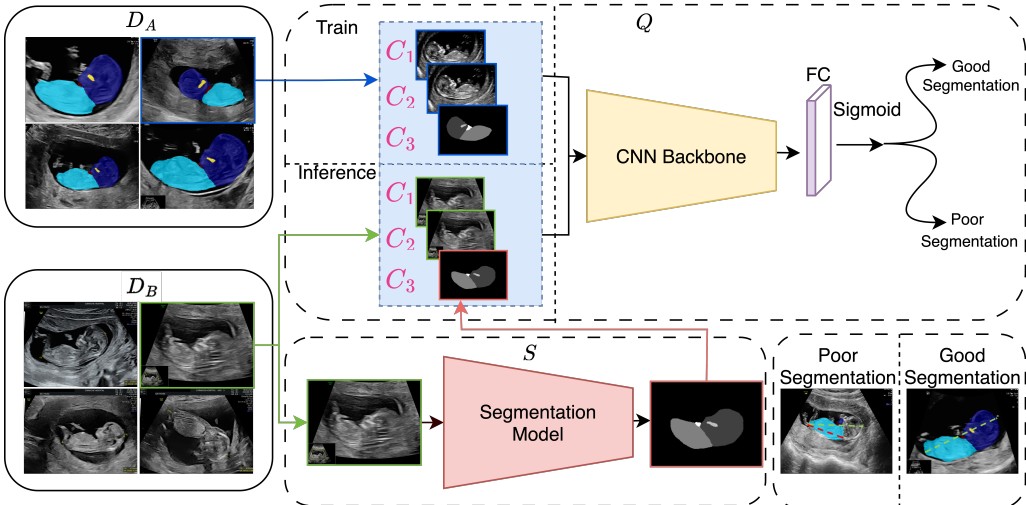

Figure 1: The train and inference pipelines for our proposed FUSQA method using two datasets $D_A$ and $D_B$. The bottom-right shows samples of good and bad segmentation masks and the impact on the CRL estimation.

Assuming we have a new set of ultrasound images $\{x_1, x_2, \ldots, x_m\} \in D_B$ without ground truth masks, we observe a deterioration of performance for $S_A$, which manifests in the deterioration in the prediction of $\hat{g}$. Normally, with the presence of ground truth masks, we would fine-tune the model on the new samples from $D_B$ to improve the model performance on the new domain. However, the process of annotating segmentation masks for new samples is time-consuming and expensive. Therefore, in this work, we propose a segmentation quality assessment model $Q$, which is used to classify whether an ultrasound image and a segmentation mask pair $(x, \hat{y})$ is of good quality for $\hat{g}$ estimation. In such a manner, our aim is for $Q$ to be trained on samples from $D_A$ and still perform well on $D_B$. Our hypothesis is that the segmentation quality assessment task is easier to transfer across datasets compared to the segmentation task.

$Q$ can be formulated as: (a) a Siamese network with two CNNs that have identical architectures and share the same weights. One of the CNNs takes $x$ as input and the other takes $\hat{y}$ as input, while their latent representations are then concatenated and fed into a fully-connected layer, the output of which is passed through a Sigmoid function that provides a probability of $\hat{y}$ being a good segmentation mask to estimate $\hat{g}$. (b) A Synergic model, which is similar to the Siamese one, but where the two CNNs do not share weights. (c) A single CNN model, the channels of the input to which is formulated as follows $(x, x, \hat{y})$ and the output is also a probability of $\hat{y}$ being a good segmentation mask to estimate $\hat{g}$. The overall outline of the method when using a single CNN is summarized in Figure 1.

Training the FUSQA model in a supervised manner requires ground truth segmentation masks. Therefore, we generate a set of altered masks $\{y'_1, y'_2, \ldots, y'_k\}$ from $y$ for every $(x, y)$ pair. To ensure stable training and avoid class imbalance, we generate $\frac{k}{2}$ poor quality masks, while the remainder $\frac{k}{2}$ are of good quality, which includes the ground truth mask $y$.

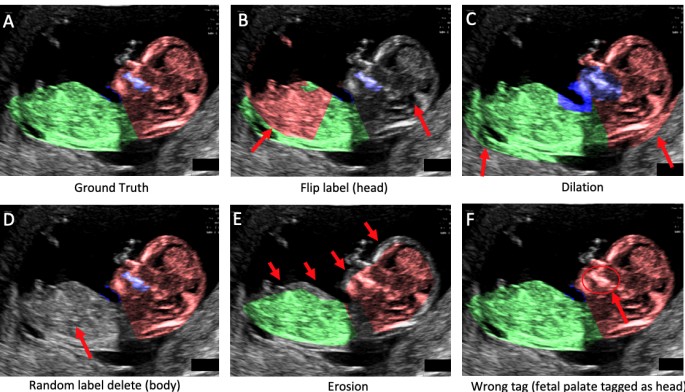

Figure 2: An example ground truth segmentation mask (A) and altered poor segmentation masks from it with the following variants: flipping of the head (B), segmentation over-estimation (C), randomly selected label deletion (D), segmentation under-estimation (E), fetal palate tagged as the head (F).

Poor segmentation masks (Figure 2) are generated using the following approaches: flipping the mask of the fetal head, segmentation dilation, randomly deleting a whole segmentation class label, segmentation erosion, and changing the fetal palate class label to a head class label. As for good quality masks, they are generated using a small amount of dilation and erosion on the segmentation mask. More details about the good and poor mask generation processes are available in the supplementary material.

## 4. Dataset

We conducted a multi-site study by collecting two different fetal ultrasound datasets acquired from a mixture of machines including Voluson S10 Expert, E8, E10, and Vivid 270 at two hospitals. The first dataset ($D_A$), acquired in United Kingdom (UK), includes 696 2D fetal ultrasound images and masks, which were segmented by an expert into four different structures: body, head, the gap between the chin and chest, and the fetal palate. The second dataset ($D_B$), which was acquired in United Arab Emirates (UAE), consists of 226 2D fetal ultrasound images acquired from a different hospital. While the first dataset contains segmentation masks, the second dataset does not. The first dataset is used for the training and validation of the segmentation and FUSQA models, while the second dataset is used for testing. The use of the second dataset as an independent test set aims to provide a way to adequately assess the quality of the automatic segmentation from an unseen dataset without the need of having manual masks. Images from both datasets were resized to a standard size of $224 \times 224$.

## 5. Experimental Settings

In this section, we describe the details surrounding the model architectures along with the training processes for different settings. First, we generate altered mask samples (Details in

Section 3) from dataset $D_A$ to train our segmentation quality assessment model, where in addition to the original ground truth mask, which is considered as good quality, we generate 4 other good quality masks and 5 poor quality masks per image and segmentation pair. We split dataset $D_A$ at the patient level into 90% train and 10% validation subsets. We use different CNN backbones (ResNet, AlexNet, VGG, and DenseNet) with different depths for the different topologies of our model (Single, Synergic, and Siamese). All models are trained by minimizing the cross-entropy loss, with the specific hyper-parameters mentioned in the supplementary material. During training, we apply data augmentation through on-the-fly random scaling, random horizontal flipping ($p = 0.5$), and rotation (between $-10°$ and $10°$) for the input images and masks. The PyTorch implementation is available at (https://github.com/BioMedIA-MBZUAI/FUSQA).

We also train Fetal-TransUnet (Cengiz et al., 2022), a segmentation model developed for fetal ultrasound segmentation, on dataset $D_A$. We follow the training strategy from (Cengiz et al., 2022), where we train the model in a 3-fold cross-validation manner, and we pick the best-performing model. This model is then used to predict segmentation masks for images from the unseen dataset $D_B$, which is used later for testing.

## 6. Results

In this section, we demonstrate the performance of our segmentation quality assessment model against expert-labeled segmentation masks against clinical guidelines. We also demonstrate the impact of our segmentation quality assessment on the gestational age downstream task. All results are reported on dataset $D_B$, which was not used in any part of the training process for both the segmentation quality assessment model and the Fetal-TransUnet model. We also compare our results on $D_B$ with our implementation of (Galati and Zuluaga, 2021), where we adapt the provided code to train the CAE on $D_A$ and we test on $D_B$. Based on the output reconstruction from the CAE, we compute a different ratio between the reconstruction and Fetal-TransUnet prediction. Then using a set threshold ($\tau$) parameter for the difference ratio, we can employ the CAE to differentiate between good ($< \tau$) and poor-quality segmentation masks. We experimentally choose a threshold of $\tau = 10\%$ difference, which results in the best classification performance.

**Classification.** We use Fetal-TransUnet to generate prediction masks from $D_B$. The predicted masks were then reviewed by an expert and classified as either good or poor-quality segmentation masks. Stratified random sampling was applied to acquire a balanced test set of 226 images with 113 poor quality masks and 113 good quality ones. This process involved checking the prediction masks for their adherence to the Fetal Anomaly Screening Programme (FASP) guideline (NHS-Screening Programmes). A predicted mask was considered as a good quality one if it resulted in a CRL that fulfills 4 or more of the 7 criteria from FASP. Table 1 summarizes the classification results of the different configurations of the segmentation quality assessment model. The highest $F_1$ score and accuracy are achieved by the simplified CNN model. The highest $F_1$ score of 0.886 was observed with ResNet 18, while the highest accuracy of 0.902 was achieved through the ResNet 50 backbone. The ResNet backbone is the best performing for the Synergic and Siamese topologies (Detailed results in the Appendix).

Table 1: Test results comparison on unseen $D_B$ between (Galati and Zuluaga, 2021), the Siamese, Synergic, and Single CNN models with different backbones.

| Model | Network | Precision | Recall | Accuracy | $F_1$ score |
|---|---|---|---|---|---|
| Siamese | ResNet 18 | 0.673 | 0.97 | 0.75 | 0.795 |
| | ResNet 50 | 0.5 | 1.0 | 0.5 | 0.66 |
| Synergic | ResNet 18 | 0.525 | 0.983 | 0.547 | 0.685 |
| | ResNet 50 | 0.746 | 0.87 | 0.787 | 0.803 |
| Proposed | ResNet 18 | 0.795 | 1.0 | 0.871 | **0.886** |
| | ResNet 50 | 0.804 | 0.982 | **0.902** | 0.87 |
| Anomaly Detection (Galati and Zuluaga, 2021) | CAE | **1.0** | 0.72 | 0.70 | 0.83 |

Table 2: The mean CRL and GA estimation errors in predicted good and poor quality segmentation masks on unseen images from $(D_B)$ (Top). A breakdown of the compliance of predicted good and poor segmentation masks with FASP criteria on the dataset $D_B$ (Bottom).

| | | Poor Seg. | Good Seg. |
|---|---|---|---|
| **Clinical downstream tasks based on proposed model** | CRL diff (mm) | 13.63 | 2.64 |
| | GA diff (days) | 7.73 | 1.45 |
| Clinical downstream tasks based on the CAE (Galati and Zuluaga, 2021) | CRL diff (mm) | 17.71 | 5.01 |
| | GA diff (days) | 10.06 | 2.79 |
| **Image Guidance Criteria** | Neutral position | 67.6% | 79.7% |
| | Fetal palate | 64.9% | 93.6% |
| | Magnification | 75% | 99.1% |
| | Fetal face direction | 76% | 99.1% |
| | Horizontal orientation | 67.6% | 100% |
| | Left caliper definition | 45.4% | 85.2% |
| | Right caliper definition | 63.9% | 95.4% |
| | Acceptance of CRL | 64.9% | 96.3% |

**Gestational age estimation.** We further evaluate the segmentation quality assessment model's performance by comparing the CRL measurement and gestational age estimation errors in predicted good-quality images against poor-quality ones. Table 2 summarizes these results based on the classification results from the most accurate segmentation quality assessment model. It is clear that the predicted good segmentation masks are able to provide a much better CRL measurement and gestational age estimation (2.64 mm and 1.45 days respectively) compared to the predicted poor ones (13.64 mm and 7.73 days respectively). The CRL and gestational age estimation errors in predicted poor masks are quite high and cannot be used for clinical reporting. The contrary applies to the predicted good masks, with error rates within acceptable ranges. This result also demonstrates the sensitivity of the segmentation model to unseen data, where clinical viability would be hindered without quality assessment mechanisms in place. Furthermore, Table 2 compares the performance between good and poor predicted masks on each criterion in the FASP guideline. There is a clear difference in accuracy between the good and poor predicted masks, where we even see a 100% accuracy for the horizontal orientation in well-predicted masks.

## 7. Discussion

The higher performance achieved by the simplified single CNN model pertains to better-determining segmentation quality based on the input image and segmentation mask. This

is also an indicator of the model's ability to generalize beyond dataset $D_A$, when compared to the Synergic, Siamese, and CAE models. Adding the segmentation mask as a channel to the CNN input dedicates convolutional filters to learn visual queues that would allow to the model to determine the segmentation quality. The Siamese networks on the other hand share weights, which could result in confusion between the visual queues learn from the input image and its corresponding segmentation mask. As for the Synergic model, it seems that the fusion of features in the latent representation space does not allow the model to learn the task as effectively. We hypothesize that this is due to better interaction in earlier layers of the single CNN model compared to the later interaction in the Synergic one. Since most proposed CNN models use 3 channels, we ensured that each network uses 3 channels. This shall make it easier to plug and play different CNNs within our proposed quality assessment model. This also allows for the use of pre-trained models on large datasets such as Imagenet. Finally, The CAE demonstrates that it can learn a good estimation of good fetal segmentation masks. However, as it does not incorporate the original ultrasound image in its input, it does not seem to generalize as well as our proposed method.

The difference between the adherence of predicted good and poor segmentation masks to the FASP guidelines (Table 2) further highlights the clinical significance of the segmentation quality assessment process. More specifically, the predicted poor segmentation masks do not exceed the 76% accuracy mark, whereas the predicted good segmentation masks mostly achieve more than 90%. The exception to this is the neutral position criterion, which depends on the challenging task of locating the fetal gap between the chin and chest. In the first trimester, the accuracy of the GA estimation based on CRL is placed between 5 and 7 days (Com, 2017). Therefore, the GA estimation based on predicted good quality segmentation masks falls well within this range at 1.45 days, while the error in predicted poor quality segmentation masks exceed this range with 7.73 days.

## 8. Conclusion

In this work, we proposed a simplified segmentation quality assessment model (FUSQA) to automatically assess the segmentation quality of a fetal ultrasound segmentation model on unseen data. This was done to address the performance deterioration problem with segmentation models as they encounter data from a new domain. We formulated the problem as a classification task to distinguish between good or poor-quality predicted masks. This formulation allowed us to effectively identify prediction masks that would result in worse CRL measurement and consequently an erroneous gestational age estimation. We validated the performance of our approach on two datasets collected from two hospitals with varying ultrasound machines. As such, our model was able to distinguish between good and poor quality segmentation masks with over 90% accuracy. Segmentation masks predicted as good lead to a mean gestational age estimation error of 1.45 days compared to 7.73 days from segmentation masks identified as poor quality.

A limitation of this work is that the segmentation quality assessment model is trained on a specific set of segmentation mask alteration methods. This can make the model susceptible to unseen types of segmentation errors. In the future, this work can be extended to train the segmentation quality assessment model in an adversarial manner including an Auto encoder model trained using Galati and Zuluaga's approach (Galati and Zuluaga, 2021), which can

lead to better generalization. Furthermore, more extensive testing on other larger datasets can further validate the efficacy and robustness of our approach.

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

## Appendix A. Detailed results

Table 3: Test results comparison on unseen DB between (Galati and Zuluaga, 2021), the Siamese, Synergic, and Single CNN models with different backbones.

| Model | Network | Pretrained | Precision | Recall | Accuracy | $F_1$ score |
|---|---|---|---|---|---|---|
| Siamese | ResNet 18 | x | 0.618 | 0.933 | 0.679 | 0.744 |
| | | ✓ | 0.673 | 0.97 | 0.75 | 0.795 |
| | ResNet 50 | x | 0.773 | 0.825 | 0.791 | 0.798 |
| | | ✓ | 0.5 | 1.0 | 0.5 | 0.66 |
| Synergic | ResNet 18 | x | 0.649 | 0.912 | 0.710 | 0.759 |
| | | ✓ | 0.525 | 0.983 | 0.547 | 0.685 |
| | ResNet 50 | x | 0.7 | 0.895 | 0.756 | 0.786 |
| | | ✓ | 0.746 | 0.87 | 0.787 | 0.803 |
| Proposed | ResNet 18 | x | 0.672 | 1.0 | 0.7124 | 0.804 |
| | | ✓ | 0.795 | 1.0 | 0.871 | **0.886** |
| | ResNet 50 | x | 0.625 | 0.991 | 0.699 | 0.767 |
| | | ✓ | 0.804 | 0.982 | **0.902** | 0.87 |
| | DenseNet 121 | x | 0.672 | 1.0 | 0.8 | 0.804 |
| | | ✓ | 0.773 | 1.0 | 0.898 | 0.872 |
| | AlexNet | x | 0.662 | 0.938 | 0.752 | 0.776 |
| | | ✓ | 0.733 | 1.0 | 0.862 | 0.846 |
| | VGG 16 | x | 0.697 | 1.0 | 0.783 | 0.821 |
| | | ✓ | 0.816 | 0.946 | 0.893 | 0.877 |
| Anomaly Detection (Galati and Zuluaga, 2021) | CAE | | 1.0 | 0.72 | 0.70 | 0.83 |

## Appendix B. Siamese and Synergic Model Architectures

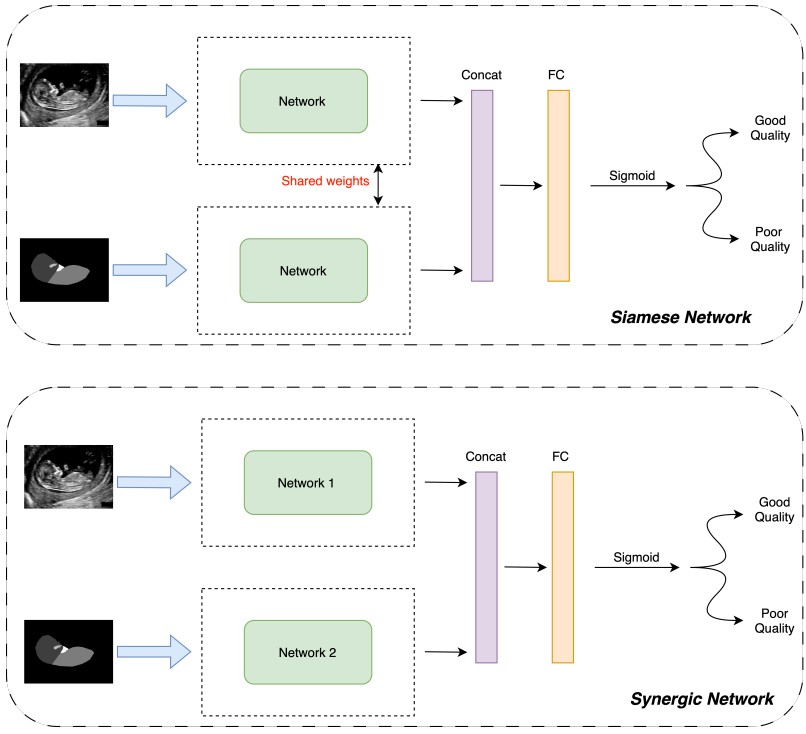

Figure 3: Siamese and Synergic networks.

## Appendix C. Step by step: generating high and poor quality image segmentation masks

We followed these pipelines to generate new segmentation masks for the poor quality:

- **Dilation:** Over-dilation was applied on the masks.

- **Erosion:** Over-erosion was applied on the masks.

- **Wrong class label:** The label number was tagged with a different class label instead of the correct label number.

- **Delete class label:** Randomly selected label was deleted.

- **Flipping the randomly selected label:** Vertical mirror of a randomly selected class while keeping the rest of classes as it is.

All these items were applied to each ultrasound image in the training and testing dataset. As a result, we have 5 different badly segmented masks of each ultrasound image. We followed these pipelines to generate new segmentation masks for good quality:

- **Dilated-segmentation:** Dilation was only applied to a randomly selected label on the masks with kernel size 3x3. The rest structures of the fetus remain the same. Ex: Dilation was applied to the fetal palate and the fetal head, body, and gap are the same.

- **Eroded-segmentation:** Erosion was applied only to a randomly selected label on the masks with kernel size 3x3. The rest structures of the fetus remain the same.

We generated 4 different good segmentation masks. As a result, we have a total of 5 good segmented images counting the original segmentation mask for each ultrasound image.

