# OpenReview forum: "FUSQA: Fetal Ultrasound Segmentation Quality Assessment"
_MIDL.io/2023/Conference — MIDL 2023 Poster_

### Official Review · Reviewer_Zqj8 · 2023-01-29

**Confidence:** 5
**Preliminary Rating:** 3
**Recommendation:** Poster

**Summary:**

The key idea of this paper is to develop a model that can accurately assess the quality of fetal ultrasound segmentation masks. This model, called FUSQA (Fetal Ultrasound Segmentation Quality Assessment), was tested on two different datasets from two hospitals and achieved over 90% accuracy in distinguishing between good and poor quality segmentations. The experiments conducted showed that when using well-segmented masks, there was only a 1.45 day difference between the gestational age reported by doctors and estimated based on CRL measurements; however, when using poorly segmented masks this difference increased up to 7.73 days which highlights how important accurate CRL measurement is for detecting potential fetal anomalies.

**Strengths:**

The strengths of this paper are that it proposes a novel approach to fetal ultrasound segmentation quality assessment, which is an important step in accurately assessing fetal growth. The proposed model was tested on two different datasets from two hospitals and achieved over 90% accuracy in distinguishing between good and poor quality segmentations. Additionally, the experiments conducted showed that when using well-segmented masks there was only a 1.45 day difference between the gestational age reported by doctors and estimated based on CRL measurements; however, when using poorly segmented masks this difference increased up to 7.73 days which highlights how important accurate CRL measurement is for detecting potential fetal anomalies.

**Weaknesses:**

- One of the weaknesses of this paper is that it does not provide a detailed comparison between its proposed model and existing approaches for fetal ultrasound segmentation quality assessment and uncertainty-based selective prediction methods.
- Additionally, while the experiments conducted showed promising results in terms of accuracy, more extensive testing on larger datasets would be needed to further validate the performance and robustness of this approach.
- It seems like FUSQA is similar to FUIQA (Wu L, Cheng JZ, Li S, Lei B, Wang T, Ni D. FUIQA: fetal ultrasound image quality assessment with deep convolutional networks. IEEE transactions on cybernetics. 2017 Mar 9;47(5):1336-49.) but for a different application. Differences should be clearly discussed in the paper.
- classifying into good and bad segmentation is quite binary and informed by a human observer. Wouldn't it be more beneficial to estimate suitability for a downstream model instead and/or to calibrate a real valued model prediction confidence metric instead?
- A conclusive experiment how the new information gain can be used is missing. I would suggest to conduct experiments with selective prediction and check which method refuses to make a prediction on the least number of test samples while keeping up excellent performance for different quality expectation thresholds.

**Deanonymize Review:**

no

**Detailed Comments:**

suggestions for further literature:
- Zhao H, Zheng Q, Teng C, Yasrab R, Drukker L, Papageorghiou AT, Noble JA. Towards Unsupervised Ultrasound Video Clinical Quality Assessment with Multi-modality Data. InMedical Image Computing and Computer Assisted Intervention–MICCAI 2022: 25th International Conference, Singapore, September 18–22, 2022, Proceedings, Part IV 2022 Sep 16 (pp. 228-237). Cham: Springer Nature Switzerland.
- Budd S, Sinclair M, Khanal B, Matthew J, Lloyd D, Gomez A, Toussaint N, Robinson EC, Kainz B. Confident head circumference measurement from ultrasound with real-time feedback for sonographers. InMedical Image Computing and Computer Assisted Intervention–MICCAI 2019: 22nd International Conference, Shenzhen, China, October 13–17, 2019, Proceedings, Part IV 22 2019 (pp. 683-691). Springer International Publishing.

**Paper Type:**

validation/application paper

**Questions To Address In The Rebuttal:**

I would like the authors to address how their proposed model compares to existing approaches for fetal ultrasound segmentation quality assessment, as well as provide details about how this AI-based system could potentially be implemented or deployed in real-world clinical settings. Additionally, I would like them to explain why further testing on larger datasets is needed and what steps they plan on taking in order to validate the performance and robustness of this approach and I would like to see a comparison between FUSQA and FUIQA.

---

### Official Review · Reviewer_mBko · 2023-02-02

**Confidence:** 5
**Preliminary Rating:** 4
**Recommendation:** Poster

**Summary:**

This paper presents a CNN to assess the quality of a segmentation previously obtained from a separate neural architecture. The resulting segmentation is classified into two categories: good/poor. The framework is applied in the context of fetal segmentation from ultrasound images, which is one of the novelties of the work.

Experiments are performed on two different datasets acquired from different machines. The paper assesses the accuracy of the classification task (quality of images) and it studies how the gestational age estimation performs as a function of the segmentation quality.

**Strengths:**

* Novel field of application (fetal image segmentation from US) for segmentation quality control
* Simple yet performing architecture
* The paper is very clear, well written and easy to follow
* Experiments include different forms of backbone architecture to demonstrate the performance of the method in different setups

**Weaknesses:**

* Segmentation quality control has been widely addressed in the literature, although it is little discussed in the paper (see detailed comments)
*The procedure to generate poor quality masks is very limiting and results in unrealistic poor quality masks (e.g. flipped skull). This aspect has been previously discussed in the literature
* Comparison is not done against methods specifically developed for medical image segmentation quality control
* The datasets are not well documented

**Deanonymize Review:**

no

**Detailed Comments:**

- There is a large body of the literature addressing quality control for medical image segmentation. Approaches have tackled it both as a regression and a classification problem (see [1-6] as an example, but there are many more). Although these have addressed different applications, from a methodological perspective, these works are equally relevant, even more as the proposed method does not encode anything that is application specific. The authors should position their work w.r.t. this literature and acknowledge it.
- The problem of the lack of representative poor mask used at training has been previously discussed (for instance by Robinson, which is cited on the paper) and specifically tackled by [1] that proposes an unsupervised approach. Please position your work in that respect
- The formulation of the estimation of the gestational age seems unnecessary,as this is not the main focus of this work and could be omitted.
- The paper should consider some of the provided references to perform a fair comparison against the SOTA. The methods used have not been specifically designed for the task (medical image segmentation QC).


[1] Galati, F et al. "Efficient model monitoring for quality control in cardiac image segmentation." Functional Imaging and Modeling of the Heart: 11th International Conference, FIMH 2021, 2021
[2] Kohlberger, T., et al: Evaluating segmentation error without ground truth. In: Medical Image Computing and Computer-Assisted Intervention. pp. 528–536 (2012)
[3] Puyol-Antón E,et al.: Automated quantification of myocardial tissue characteristics from native t1 mapping using neural networks with uncertainty-based quality-control. Journal of Cardiovascular Magnetic Resonance 22(1) (2020)
[4] Ruijsink, B., et al.: Fully automated, quality-controlled cardiac analysis from CMR. JACC: Cardiovascular Imaging 13(3), 684–695 (2020)
[5] Sunoqrot, M, et al. "A quality control system for automated prostate segmentation on T2-weighted MRI." Diagnostics 10.9 (2020): 714.
[6] Gonzalez, C et al. "Self-supervised out-of-distribution detection for cardiac CMR segmentation." Medical Imaging with Deep Learning 2021

**Justification of score**
I appreciate the efforts made by the authors to improve their manuscript by addressing the points raised by the reviewers and conducting further experiments. Quality control of US images is a topic that is less explored than in other modalities (e.g. CT or MRI). Therefore I believe the paper may bring new elements for discussion to the conference and I support its acceptance.

**Paper Type:**

methodological development

**Questions To Address In The Rebuttal:**

Following the points raised in the detailed comments section, please
1) position your work w.r.t the previous literature
2) position the limitations of your work, regarding mask generation, w.r.t. previous works (see ref [1] in detailed comments) that have proposed solutions to it
3) if time permits, show results of other competing methods that have been designed for QC in medical image segmentation QC.

---

### Official Review · Reviewer_rKmm · 2023-02-04

**Confidence:** 4
**Preliminary Rating:** 3
**Recommendation:** Poster

**Summary:**

This manuscript proposes a framework to assess the quality of fetal segmentation in the perspective of the prediction of gestational age. The developed model is applied to an unseen dataset and demonstrate that what they classify as poor segmentation quality result in larger error for clinical dowstream tasks and image guidance criteria

**Strengths:**

The comparison to other architectures is well led in the experiment section as it allows to observe the architecture dependency on the final results
One important strength of this study is the importance put on the downstream clinically relevant metrics (error in gestational age prediction and findings of image guidance)

**Weaknesses:**

The main weakness of this paper lies in what seems a missing opportunity in the paper. While the authors argue that one of the issues in DL based methods for fetal segmentation lies in the lack of generalisability, their proposed method only addresses the problem of assessment of quality in a secondary dataset but don't really provide any suggestion on how to solve it. They mention at time that what they propose may help in improving generalisability of segmentation but there is no proof about it.

At times, lack of clarity prevents the message of the paper to be straightforward (e.g creation of 5 good quality segmentations)
This lack of clarity pertains also to the captions of the tables that are impossible to understand as standalone from the text

**Deanonymize Review:**

no

**Detailed Comments:**

One example of need for rephrasing at times is in p6 "we generate 9 masks in total 5 poor quality and 5 good quality ones including the ground truth" should probably be modified as overall in addition to the original ground truth considered as good quality we generated 4 other good quality masks and 5 poor quality masks per image pair.

One example for the lack of clarity of the captions relates to Table 1 where we don't know in the caption on which dataset this has been calculated, how many poor and good quality elements are supposed to be assessed...



**Paper Type:**

methodological development

**Questions To Address In The Rebuttal:**

Please do focus particularly on the question of the use for segmentation improvement as this is essential to really assess the motivation and clinical relevance of the paper.
Clarification of the text and completion of the tables captions should be modified for final submission.

---

### Meta-Review · Area_Chair_mP3x · 2023-02-22

**Recommendation:** Accept (Poster)
**Confidence:** 5

**Metareview:**

Interesting use case: automatic detection of segmentation quality in unseen images. It could be used for quality control and detection of errors by a deep learning algorithm. The performance of quality control is evaluated using a downstream task, i.e., gestational age estimation. The authors perform experiments using data sets from two centers and find that the trained model is able to identify poor segmentations in the unseen data set. This is a clear application paper that could have an impact on the automatic processing of (large) sets of fetal ultrasounds and might find applications in the clinic. The reviewers have raised several issues and started a discussion in which the authors have engaged. Among other changes, authors have substantially improved their related works section. This paper would be a nice fit for MIDL.